# An exploratory study on the handwritten allographic features of multi-ethnic population with different educational backgrounds

Linthini Gannetion[1☯], Kong Yong Wong[2☯], Poh Ying Lim[3☯], Kah Haw Chang[1☯], Ahmad Fahmi Lim Abdullah[1☯]*

**1** Forensic Science Programme, School of Health Sciences, Universiti Sains Malaysia, Kubang Kerian, Kelantan, Malaysia, **2** Document Examination Division, Forensic Science Analysis Centre, Department of Chemistry Malaysia, Jalan Sultan, Petaling Jaya, Selangor, Malaysia, **3** Faculty of Medicine and Health Sciences, Department of Community Health, Universiti Putra Malaysia, Serdang, Selangor, Malaysia

☯ These authors contributed equally to this work.
* fahmilim@usm.my

**Data Availability Statement:** All relevant data are within the paper and its Supporting Information files.

## Abstract

Ethnicity, native and/or foreign language knowledge, as well as the learned writing systems potentially influence the development of an individual's handwriting. The unique education system consisting of National schools, Chinese-medium vernacular schools, Tamil-medium vernacular schools, and Islamic religious schools in Malaysia may have established specific characteristic handwritten allographic features that deserve investigation within the intelligence context. This study was aimed to explore handwritten allographic features of handwriting samples from 120 subjects (30 writers from four different educational backgrounds mentioned above). Characteristic features which could be attributed to the study groups were statistically analyzed and identified. In this study, thirteen allographic features, including letters "A", "B", "D", "H", "p", "T", "t", "w", "X" and "x", were found to be discriminative. Such information could serve to indicate the primary education system undergone by a writer; enabling the comparison of different handwriting profiles and allowing characterization of writers to a specific group of people.

## Introduction

Ethnicity, native and/or foreign language knowledge, and writing systems are factors that contribute to the development of one's handwriting [1]. The possibility to detect habitual features in English handwritings to indicate culture could be due to the varying attributes of writing systems in western countries [2]. In the west, special letter shapes are retained in the handwritings of people who learned writing in Canada and the Netherlands, which are influenced by the copybook systems in respective countries [3]. Influences of taught writing styles were also observed in the handwritings between Polish and English [4], Indian and English [5, 6], as well as Vietnamese and Australian [7] writers in English written documents. However, only 1% of

**Funding:** The authors thank the financial support via the Fundamental Research Grant Scheme (FRGS) from Ministry of Higher Education Malaysia (FRGS/1/2018/SS10/USM/02/1, awarded to Ahmad Fahmi Lim bin Abdullah). The funder had no role in study design, data collection and analysis, decision to publish, or preparation of the manuscript.

**Competing interests:** The authors have declared that no competing interests exist.

a population would continue to adhere to the copybook writing style through the examination of grapheme "th" [8]. Identification of a writer's nationality based on copybook system was also found not successful, restricted by abundant variations in handwriting characteristics in certain countries [9]. Handwritings from England in English written documents were most likely to be confused with styles from other countries due to lack of emphasis on copybooks [9].

Compared to studies between people originating from different countries of varying native language knowledge, studies on handwriting class characteristics for individuals from a single country are limited. From the available literature, Cheng and co-researchers had identified six class characteristics of handwritings to characterize three major races in Singapore [10]. They suggested that the majority of Indians in Singapore have the habit of exhibiting curve stroke formation in the letter "X" and round top formation in the letter "A" while most Chinese in the country write the horizontal stroke first when writing letter "T". Additionally, the looping of "d" stem, looping on "f" lower stem, and straight stroke formation of letter 'M' and 'N' were found possible in differentiating different racial groups [10].

Writers of different nationalities, native language knowledge and the learned writing systems could have exhibited discriminative handwriting characteristics [6, 10]. Applicability of previous studies to Malaysian's handwriting is unknown simply because the unique education system today was established during the British administration before Malaysia gained independence in 1957. Malaysia's education system consists of four different primary educational systems based on the language or medium of instruction used in teaching and learning, namely: Malay-medium National schools, vernacular schools of Chinese and Tamil media, and Islamic religious schools. In addition to Malay and English languages as compulsory subjects in all schools, students also learn Chinese, Tamil, and Arabic languages in their respective schools. Hence, handwritten allographic features adapted during primary levels at different educational systems which is the copybook writing style, and their influence towards the formation of one's handwriting known as class characteristics is of great interest for forensic intelligence pertaining to handwritten evidence. Specifically, knowledge on the influence of an individual's educational background towards the development of handwriting could provide additional information in forensic handwriting examination.

Forensic handwriting examination which is part of questioned document examination (QDE) aims to identify the author of a questioned handwriting using defensible scientific methods and to determine whether the author has produced a series of documents or if the signature was falsified [1, 2]. In most cases, QDE involves the comparison of an unknown writing sample with samples from a known writer to establish similarities and/or differences between the writing of the two sources [1, 2, 11, 12]. Such comparison is mandatory during forensic investigation in criminal procedures or civil disputes, but it is only possible when comparable writing samples are available to forensic document examiners. In cases where there was no known suspect or victim or in the absence of comparable handwriting samples (such as an untraceable parcel sent to an individual or a written document received from an unknown sender), the application of conventional QDE comparison using known and suspect samples becomes impossible. In this situation, comparing the profiles of questioned handwriting to a group or database of handwriting can help narrow down the pool potential writers in document related fraud, forensic accounting and even in terrorism investigation [11].

The premise here is that the variations in academic backgrounds, or more specifically, the educational learning systems by Malaysians are believed to have influenced their handwriting formation and this remains unexplored. This study was aimed to identify and establish the characteristic handwritten allographic features corresponding to selected population samples from different primary educational backgrounds. It is hoped that the determination of these

features coupled with examinations of each allographic feature would serve as initial data, enabling the comparison of different handwriting profiles within the intelligence context and allow for characterization of writers to a specific group.

## Materials and methods

### Study population

In this study, each member of specific sample populations (Malay-medium National schools, vernacular schools of Chinese and Tamil mediums and Islamic religious schools) had an equal chance of being included. The writer was chosen by chance for actual inclusion. A name list of university students was obtained upon institutional approval and a computer generated simple random sampling method was used to find potential participants. An email was sent to the potential writers with the details of the study, including the purpose of study, risk of providing handwriting samples, and the disposal of handwriting samples upon completion of the study. The determined venue and time for handwriting sample collection were given to participants, and their consents were obtained. Sample size was calculated through two proportion formula by considering the letter "U" with introductory stroke [4], 25% of the Polish (0% for English) possessed the handwritten feature. With 80% of power study, 5% of risk error, and 10% of non-response rate, 30 writers were collected from each group. This resulted in a total number of 120 writers who voluntarily participated in the current study. Only writers aged between 18 to 20 years old and well-versed in reading and writing in English Language were recruited. A qualification level of higher than pre-university level with no mental and physical disabilities were also considered as inclusion criteria for the writers. Ethical approval was obtained from the institutional Human Research Ethics Committee (USM/JEPeM/17120687).

### Data collection

A data collection kit comprising of an information sheet, consent form, writer's information form, a source document and three blank sheets, as well as a blue ballpoint pen were distributed to each writer. Information pertaining to the age, gender, race, handedness, occupation, education level, primary and secondary schooling systems were specified in the writer's information form prior to collecting the handwriting samples. The writers were required to copy a source document, thrice, on blank A4 white papers using the blue ballpoint pen provided in the kit. The source document is an English text, similarly used by Srihari *et al.* [13]. The source text consists of 156 words capturing all letters and numerals, punctuation as well as distinctive letter and numeral combinations (graphemes of ff, tt, oo, and 00).

### Data extraction and coding

Background data was organized in a Microsoft Excel® Spreadsheet (Redmond, WA). Individual spreadsheets were created for individual alphabets and a separate spreadsheet was created to record the writers' background data, age, handedness, education level, and their respective learning systems. Writers with Malay-medium National schools background were designated as Group A whereas writers from Chinese medium and Tamil medium vernacular schools as well as the Islamic religious schools were designated as Groups B, C, and D, respectively.

All handwriting samples were scanned at 400 dot per inch (dpi) using Canon® Image-CLASS MF3010 scanner (Canon Inc., Ota City, Tokyo, Japan) and viewed at magnification 2:1 using Adobe® Photoshop® CC software (Adobe Inc., San Jose, CA) for detailed examination of the handwriting features. Allographic features of each alphabet were examined by investigating the (1) number of strokes, (2) sequence of strokes, (3) direction of crossbar stroke, (4)

position of crossbar strokes relative to the average letter height, (5) connectivity of crossbar stroke to the adjacent letter, (6) length of crossbar strokes, (7) connectivity between strokes, (8) morphology of individual strokes, (9) apex design and/or (10) presence of hiatus and looping. Each said feature was assigned with a code value.

In this study, the number of writers with the features occurring repeatedly in their handwriting samples were counted and organized in the spreadsheet. Frequency of occurrence for each handwritten allographic feature was calculated and compared across the four groups of writers. Based on the comparison, alphabets which showed significant frequency of allographic features attributed to a specific study group were then identified.

## Statistical analysis

IBM SPSS Statistics version 27 (SPSS Inc., Chicago, IL, USA) was used for data analysis. Data cleaning and descriptive analyses were performed to ensure there were no key in errors. All identified handwritten allographic features were analyzed using Pearson's Chi-square tests to determine if there are associations between the four groups of writers from based on frequency of occurrence for each feature. A p-value <0.05 was considered statistically significant. The significant variables were treated as confounding variables and controlled in the regression model.

Multinomial logistic regression model was used to investigate each common handwritten allographic features associated with the educational backgrounds, adjusted with the confounding variables. The regression model was constructed as follows:

$$y_i = \alpha_i + \beta_1 X_i + \beta_2 X_i + \mu_i$$

$$\mu_i \sim N(0, \sigma^2)$$

where $y_i$ was a nominal variable with four categories (types of educational background), $\alpha_i$ was intercept of equation, $\beta_1$ was the coefficient of common handwritten allographic features, $\beta_2$ was the coefficient of confounding variables $\mu_i$ was random effect of respondent level, with the variation $\sigma^2$. A p-value <0.05 was considered statistically significant in the regression model. The common handwritten allographic features in one or more groups as compared to others were determined and evaluated.

# Results

## Data description

All the 120 samples collected in this study were written using the ballpoint pens provided in the kit with the aim to minimize variation which could have arisen from using different writing instruments. Both the original and scanned copies of the handwriting samples were examined to determine the characteristic allographic features by the research team which was also membered by a certified forensic document examiner of the country. During the initial stage, handwritten features of all the alphabets and numbers in each handwriting sample were examined and coded in two separate occasions independently. Subsequently, the coded results were compared and validated by the certified document examiner. In this study, only the validated data was considered for interpretation and statistical analyses.

Based on the handwriting samples, 18 allographic features were suggested as the characteristics which potentially occurred in one group as compared to other groups, covering 14 letters "A", "B", "D", "H", "I", "P", "p", "T", "t", "w", "X", "x", "Z" and "z". Four letters ("A", "B", "D", and "w") demonstrated two features to be tested at differentiating the handwritings of the four study groups. Description of the 18 allographic features is shown in Fig 1.

| Description | Allographic feature |
|---|---|
| Uppercase "A" with three individual strokes | |
| Uppercase "A" with crossbar exceeding the letter boundary | |
| Uppercase "B" with two individual strokes | |
| Uppercase "B" with no protruding initial stroke | |
| Uppercase "D" with two individual strokes | |
| Uppercase "D" with protruding initial stroke | |
| Uppercase "H" with commanding stroke written first followed by crossbar and vertical stroke | |
| Uppercase "I" written as single stroke (vertical stroke) without upper and lower serif | |
| Uppercase "P" with one continuous stroke | |
| Lowercase "p" with one continuous stroke | |
| Uppercase "T" with vertical stroke written first followed by arm | |
| Lowercase "t" with crossbar written first followed by vertical stroke | |
| Lowercase "w" with left and right sections apart | |
| Lowercase "w" with sharp left and right vertices | |
| Uppercase "X" with right to left diagonal stroke constructed first followed by left to right diagonal stroke | |
| Lowercase "x" with right to left diagonal stroke constructed first followed by left to right diagonal stroke | |
| Uppercase "Z" with rounded top arm-junction design | |
| Lowercase "z" with rounded top arm-junction design | |

**Fig 1. Description 18 allographic features suggested as the characteristic of at least one study group.**

**Table 1. Description on demographic information of the participants (N = 120).**

| Variable | | National school (n = 30) | Chinese vernacular school (n = 30) | Tamil vernacular school (n = 30) | Islamic religious school (n = 30) | p-value |
|---|---|---|---|---|---|---|
| Age | 18/19 | 27(90.0%) | 12(40.0%) | 11(36.6%) | 24(80.0%) | <0.001* |
| | 20 | 3(10.0%) | 18(60.0%) | 19(63.4%) | 6(20.0%) | |
| Gender | Male | 6(20.0%) | 8(26.7%) | 4(13.3%) | 4(13.3%) | 0.485 |
| | Female | 24(80.0%) | 22(73.3%) | 26(86.7%) | 26(86.7%) | |
| Ethnicity | Malay | 25(83.4%) | 0(0%) | 0(0%) | 29(96.7%) | <0.001* |
| | Non-Malay | 5(16.6%) | 30(100%) | 30(100%) | 1(3.3%) | |
| Handedness | Left | 6(20.0%) | 2(6.7%) | 2(6.7%) | 2(6.7%) | 0.217 |
| | Right | 24(80.0%) | 28(93.3%) | 28(93.3%) | 28(93.3%) | |
| Secondary Schooling System | National School | 30(100%) | 19(63.3%) | 29(96.7%) | 16(53.3%) | <0.001* |
| | Others | 0(0%) | 11(36.7%) | 1(3.3%) | 14(46.7%) | |

Pearson's Chi Square test was performed.

*p-value<0.05

The demographic information of the participants from the four different educational backgrounds was assessed, as tabulated in Table 1. No association was found in gender (p = 0.485) and handedness (p = 0.217) in relation to primary educational backgrounds. However, age, ethnicity, and secondary schooling system demonstrated significant association with educational backgrounds (p <0.001). Due to the large proportion of writers having similar ethnicity as in their respective educational backgrounds, two confounding variables, *i.e.* age and secondary schooling system, were adjusted in the final model through multinomial logistic regression analysis.

Frequency differences of individuals who exhibited the allographic features, and those who did not, were determined using Pearson's Chi Square test (Table 2).

There was significant association between educational backgrounds and 14 individual allographic features from a total of 18 features. Letter "t" with crossbar written first followed by vertical stroke gave the highest discriminatory power ($\chi2$ = 78.348, df = 3, p < 0.001) and was observed in more than 83% of the handwriting samples of writers from Chinese vernacular school group. By comparison, this feature was only found in approximately 13% of samples obtained from individuals from Tamil medium vernacular school background, and none of this feature was noted in the other two study groups.

Other discriminative features included the construction of letters "D" with protruding initial stroke ($\chi2$ = 27.768, df = 3, p < 0.001), both "X" ($\chi2$ = 23.893, df = 3, p < 0.001) and "x" ($\chi2$ = 22.289, df = 3, p < 0.001) with right to left diagonal stroke constructed first followed by left to right diagonal stroke, as well as letter "A" with crossbar exceeding the letter boundary ($\chi2$ = 20.124, df = 3, p < 0.001). Multinomial logistic regression analysis was subsequently carried out (Table 3 is referred). The statistical outcome was nominal variable with four categories of educational background with one of the educational backgrounds served as the reference group. Each allographic feature was treated as an independent variable.

Table 3 demonstrates the coefficients of the multinomial logistic regression model, with statistically significant coefficients highlighted. A p-value of less than 0.05 allowed for comparison between the two groups of participants in relation to their handwritten allographic features. In our case, a value of more than 1 in the odd ratio (OR) indicated a positive sign while OR less than 1 suggested a negative sign. Considering the feature of the uppercase "A" with crossbar exceeding the letter boundary, a positive sign indicated that it is more likely that the writer is

**Table 2. Association of allographic features with educational backgrounds using Pearson's Chi Square test.**

| Allographic Features | Group | | | | $\chi^2$ (df) | *p*-value |
|---|---|---|---|---|---|---|
| | National school | Chinese vernacular school | Tamil vernacular school | Islamic religious school | | |
| Uppercase "A" with three individual strokes | 17 (56.7%) | 8 (26.7%) | 10 (33.3%) | 8 (26.7%) | 7.937 (3) | **0.047**\* |
| Uppercase "A" with crossbar exceeding the letter boundary | 6 (20.0%) | 7 (23.3%) | 5 (16.6%) | 19 (63.3%) | 20.124 (3) | < **0.001**\* |
| Uppercase "B" with two individual strokes | 9 (30.0%) | 16 (53.3%) | 17 (56.7%) | 6 (20.0%) | 11.955 (3) | **0.008**\* |
| Uppercase "B" with no protruding initial stroke | 14 (46.7%) | 18 (60.0%) | 24 (80.0%) | 10 (33.3%) | 14.411 (3) | **0.002**\* |
| Uppercase "D" with two individual strokes | 12 (40.0%) | 23 (76.7%) | 21 (70.0%) | 10 (33.3%) | 16.835 (3) | **0.001**\* |
| Uppercase "D" with protruding initial stroke | 21 (70.0%) | 3 (10.0%) | 7 (23.3%) | 8 (26.7%) | 27.768 (3) | < **0.001**\* |
| Uppercase "H" with commanding stroke written first followed by crossbar and vertical stroke | 15 (50.0%) | 25 (83.3%) | 10 (33.3%) | 11 (36.7%) | 18.772 (3) | < **0.001**\* |
| Uppercase "I" written as single stroke (vertical stroke) without upper and lower serif | 16 (53.3%) | 9 (30.0%) | 8 (26.7%) | 16 (53.3%) | 7.830 (3) | 0.050 |
| Uppercase "P" with one continuous stroke | 13 (43.3%) | 10 (33.3%) | 8 (26.7%) | 15 (50.0%) | 4.089 (3) | 0.252 |
| Lowercase "p" with one continuous stroke | 16 (53.3%) | 12 (40.0%) | 9 (30.0%) | 25 (83.3%) | 19.335 (3) | < **0.001**\* |
| Uppercase "T" with vertical stroke written first followed by arm | 7 (23.3%) | 5 (16.6%) | 8 (26.7%) | 15 (50.0%) | 9.156 (3) | **0.027**\* |
| Lowercase "t" with crossbar written first followed by vertical stroke | 0 (0%) | 25 (83.3%) | 4 (13.3%) | 0 (0%) | 78.348 (3) | < **0.001**\* |
| Lowercase "w" with left and right sections apart | 10 (33.3%) | 20 (66.7%) | 13 (43.3%) | 22 (73.3%) | 12.990 (3) | **0.005**\* |
| Lowercase "w" with sharp left and right vertices | 6 (20.0%) | 10 (33.3%) | 4 (13.3%) | 18 (60.0%) | 17.175 (3) | **0.001**\* |
| Uppercase "X" with right to left diagonal stroke constructed first followed by left to right diagonal stroke | 13 (43.3%) | 11 (36.7%) | 11 (36.7%) | 27 (90.0%) | 23.893 (3) | < **0.001**\* |
| Lowercase "x" with right to left diagonal stroke constructed first followed by left to right diagonal stroke | 13 (43.3%) | 8 (26.7%) | 11 (36.7%) | 25 (83.3%) | 22.289 (3) | < **0.001**\* |
| Uppercase "Z" with rounded top arm-junction design | 17 (56.7%) | 8 (26.7%) | 9 (30.0%) | 11 (36.7%) | 6.933 (3) | 0.074 |
| Lowercase "z" with rounded top arm-junction design | 17 (56.7%) | 8 (26.7%) | 10 (33.3%) | 11 (36.7%) | 6.345 (3) | 0.096 |

from the Islamic School (Group D) than the National School (Group A- Ref group) if he/she had the feature in his/her handwriting (OR = 5.779, 95%CI: 1.571, 21.123, p-value = 0.008). On the other hand, a negative sign indicated that the writer is more likely to have originated from the Chinese Vernacular School (Group B) than the National School (Group A- Ref group) if he/she does not construct his/her uppercase "A" with three individual strokes (OR = 0.271, 95%CI: 0.077, 0.959, p-value = 0.043). The multinomial logistic regression model had allowed for the determination of specific handwritten features based on respective educational backgrounds.

This study served as an exploratory study to investigate the presence of significant differences among handwritings from individuals from the same country but with different educational backgrounds. The powers of study for each significant variable were tested, and those variable with more than 78% of power were highlighted (Table 3 is referred). This statistical

**Table 3. Probability of handwritten allographic features from writers of different educational backgrounds by multinomial logistic regression analysis.**

| Allographic Features | Group | Multinomial Logistic Regression | | |
|---|---|---|---|---|
| | | OR | 95%CI of OR | p-value |
| Uppercase "A" with three individual strokes | **A (Ref) vs B** | **0.271** | **0.077** | **0.959** | **0.043***
| | A (Ref) vs C | 0.34 | 0.103 | 1.122 | 0.077 |
| | **A (Ref) vs D** | **0.233** | **0.066** | **0.824** | **0.024***
| | B (Ref) vs C | 1.254 | 0.389 | 4.041 | 0.704 |
| | B (Ref) vs D | 0.859 | 0.253 | 2.914 | 0.808 |
| | C (Ref) vs D | 0.685 | 0.189 | 2.482 | 0.565 |
| Uppercase "A" with crossbar exceeding the letter boundary | A (Ref) vs B | 1.367 | 0.336 | 5.566 | 0.663 |
| | A (Ref) vs C | 1.006 | 0.242 | 4.184 | 0.994 |
| | **A (Ref) vs D** | **5.779** | **1.571** | **21.123** | **0.008***#
| | B (Ref) vs C | 0.736 | 0.189 | 2.861 | 0.658 |
| | **B (Ref) vs D** | **4.227** | **1.296** | **13.792** | **0.017***#
| | **C (Ref) vs D** | **5.745** | **1.512** | **21.828** | **0.010***#
| Uppercase "B" with two individual strokes | A (Ref) vs B | 0.971 | 0.267 | 3.597 | 0.976 |
| | A (Ref) vs C | 1.522 | 0.45 | 5.149 | 0.500 |
| | A (Ref) vs D | 0.26 | 0.058 | 1.159 | 0.077 |
| | B (Ref) vs C | 1.552 | 0.495 | 4.861 | 0.451 |
| | **B (Ref) vs D** | **0.265** | **0.074** | **0.947** | **0.041***#
| | **C (Ref) vs D** | **0.171** | **0.042** | **0.699** | **0.014***#
| Uppercase "B" with no protruding initial stroke | A (Ref) vs B | 0.629 | 0.177 | 2.24 | 0.474 |
| | A (Ref) vs C | 2.276 | 0.649 | 7.983 | 0.199 |
| | A (Ref) vs D | 0.316 | 0.087 | 1.149 | 0.08 |
| | B (Ref) vs C | 3.619 | 0.997 | 13.131 | 0.05 |
| | B (Ref) vs D | 0.503 | 0.155 | 1.628 | 0.251 |
| | **C (Ref) vs D** | **0.139** | **0.036** | **0.543** | **0.005***#
| Uppercase "D" with two individual strokes | A (Ref) vs B | 1.813 | 0.492 | 6.681 | 0.371 |
| | A (Ref) vs C | 1.498 | 0.441 | 5.089 | 0.517 |
| | A (Ref) vs D | 0.354 | 0.094 | 1.334 | 0.125 |
| | B (Ref) vs C | 0.827 | 0.228 | 2.99 | 0.772 |
| | **B (Ref) vs D** | **0.195** | **0.056** | **0.677** | **0.010***#
| | **C (Ref) vs D** | **0.236** | **0.062** | **0.906** | **0.035***#
| Uppercase "D" with protruding initial stroke | **A (Ref) vs B** | **0.051** | **0.01** | **0.259** | **<0.001***#
| | **A (Ref) vs C** | **0.186** | **0.052** | **0.656** | **0.009***#
| | **A (Ref) vs D** | **0.146** | **0.039** | **0.543** | **0.004***#
| | B (Ref) vs C | 3.619 | 0.726 | 18.033 | 0.117 |
| | B (Ref) vs D | 2.852 | 0.629 | 12.941 | 0.174 |
| | C (Ref) vs D | 0.788 | 0.198 | 3.138 | 0.736 |
| Uppercase "H" with commanding stroke written first followed by crossbar and vertical stroke | **A (Ref) vs B** | **5.341** | **1.326** | **21.515** | **0.018***#
| | A (Ref) vs C | 0.51 | 0.157 | 1.658 | 0.263 |
| | A (Ref) vs D | 0.504 | 0.152 | 1.669 | 0.262 |
| | **B (Ref) vs C** | **0.096** | **0.026** | **0.352** | **<0.001***#
| | **B (Ref) vs D** | **0.094** | **0.025** | **0.355** | **<0.001***#
| | C (Ref) vs D | 0.988 | 0.281 | 3.474 | 0.985 |

*(Continued)*

**Table 3.** (Continued)

| Allographic Features | Group | Multinomial Logistic Regression | | |
|---|---|---|---|---|
| | | OR | 95%CI of OR | | p-value |
| Lowercase "p" with one continuous stroke | A (Ref) vs B | 1.107 | 0.32 | 3.836 | 0.873 |
| | A (Ref) vs C | 0.662 | 0.202 | 2.164 | 0.495 |
| | **A (Ref) vs D** | **5.813** | **1.425** | **23.708** | **0.014***  |
| | B (Ref) vs C | 0.598 | 0.184 | 1.943 | 0.392 |
| | **B (Ref) vs D** | **5.25** | **1.471** | **18.746** | **0.011***# |
| | **C (Ref) vs D** | **8.783** | **2.184** | **35.316** | **0.002***# |
| Uppercase "T" with vertical stroke written first followed by arm | A (Ref) vs B | 0.755 | 0.178 | 3.201 | 0.703 |
| | A (Ref) vs C | 1.33 | 0.362 | 4.887 | 0.668 |
| | **A (Ref) vs D** | **4.307** | **1.229** | **15.09** | **0.022*** |
| | B (Ref) vs C | 1.76 | 0.471 | 6.585 | 0.401 |
| | **B (Ref) vs D** | **5.703** | **1.571** | **20.697** | **0.008***# |
| | C (Ref) vs D | 3.239 | 0.918 | 11.431 | 0.068 |
| Lowercase "t" with crossbar written first followed by vertical stroke | A (Ref) vs B | | | | - |
| | A (Ref) vs C | | | | - |
| | A (Ref) vs D | | | | - |
| | **B (Ref) vs C** | **0.015** | **0.002** | **0.101** | **<0.001***# |
| | B (Ref) vs D | | | | - |
| | C (Ref) vs D | | | | - |
| Lowercase "w" with left and right sections apart | **A (Ref) vs B** | **4.64** | **1.322** | **16.291** | **0.017*** |
| | A (Ref) vs C | 1.698 | 0.525 | 5.493 | 0.376 |
| | **A (Ref) vs D** | **5.315** | **1.523** | **18.551** | **0.009***# |
| | B (Ref) vs C | 0.366 | 0.119 | 1.123 | 0.079 |
| | B (Ref) vs D | 1.145 | 0.351 | 3.737 | 0.822 |
| | C (Ref) vs D | 3.129 | 0.917 | 10.684 | 0.069 |
| Lowercase "w" with sharp left and right vertices | A (Ref) vs B | 2.971 | 0.77 | 11.47 | 0.114 |
| | A (Ref) vs C | 0.826 | 0.187 | 0.3642 | 0.800 |
| | **A (Ref) vs D** | **7.99** | **2.518** | **29.587** | **0.002***# |
| | B (Ref) vs C | 0.278 | 0.072 | 1.076 | 0.064 |
| | B (Ref) vs D | 2.689 | 0.869 | 8.326 | 0.086 |
| | **C (Ref) vs D** | **9.677** | **2.359** | **39.694** | **0.002***# |
| Uppercase "X" with right diagonal stroke constructed first followed by left diagonal stroke | A (Ref) vs B | 0.622 | 0.175 | 2.206 | 0.462 |
| | A (Ref) vs C | 0.735 | 0.227 | 2.38 | 0.608 |
| | **A (Ref) vs D** | **10.984** | **2.381** | **50.66** | **0.002***# |
| | B (Ref) vs C | 1.182 | 0.376 | 3.716 | 0.774 |
| | **B (Ref) vs D** | **17.659** | **3.996** | **78.049** | **<0.001***# |
| | **C (Ref) vs D** | **14.937** | **3.163** | **70.546** | **0.001***# |
| Lowercase "x" right diagonal stroke constructed first followed by left diagonal stroke | A (Ref) vs B | 0.368 | 0.099 | 1.372 | 0.137 |
| | A (Ref) vs C | 0.696 | 0.214 | 2.26 | 0.546 |
| | **A (Ref) vs D** | **5.764** | **1.517** | **21.898** | **0.010***# |
| | B (Ref) vs C | 1.889 | 0.569 | 6.275 | 0.299 |
| | **B (Ref) vs D** | **15.645** | **4.055** | **60.354** | **<0.001***# |
| | **C (Ref) vs D** | **8.281** | **2.097** | **32.705** | **0.003***# |

*p-value<0.05

#Power of study ≥78%

OR = Odds Ratio; CI = Confidence Interval; Ref = Reference group; All models were adjusted with two confounding variables (age and secondary schooling system): Group A- National school; Group B- Chinese vernacular school; Group C- Tamil vernacular school; Group D- Islamic religious school.

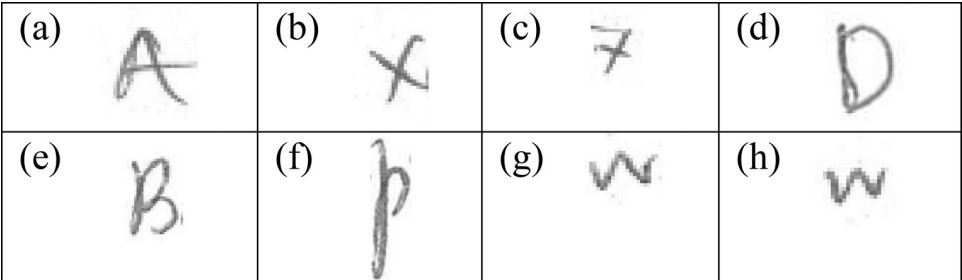

**Fig 2. Characteristic handwritten allographic features from writers from Islamic religious schools.**

power suggested that the test results were likely valid with low Type II error. Nonetheless, the handwritten allographic features should be further tested with greater number of samples. Handwritten feature with low powder of study (<70%) is suggested for further investigation with greater sample size, in this case the uppercase "A" with three individual strokes. For the current study, only those statistically significant features with adequate power (in this case greater than 78%) are discussed.

## Discussion

### Handwritten allographic features of Islamic religious school background

In general, individuals who had undergone their primary education in Islamic religious school (Group D), were more likely to carry three characteristic allographic features in their English handwriting as compared to the other groups. The three features are the: (1) uppercase "A" with crossbar exceeding the letter boundary [Fig 2(A)], (2) uppercase "X" [Fig 2(B)], and (3) lowercase "x" [Fig 2(C)] where right to left diagonal strokes were constructed first followed by left to right diagonal strokes. Islamic religious school students learn Arabic language as the main language for religion-related subjects. Unlike English, Arabic letters are written from right to left in constructing script. Shapes and construction of the script could vary based on their position in a word, whether it is written initially, medially, finally or as a separate letter; relative to its adjacent letter [14]. Arabic script is generally cursive and is usually written continuously to connect one letter to another while forming a word. In writing the uppercase "X" and lowercase "x", writers in Group D were prone to construct the right to left diagonal stroke first, most probably due to the habit adopted while learning to write Arabic scripts. The observation of significantly higher frequency of long crossbar exceeding boundary of letter "A" in

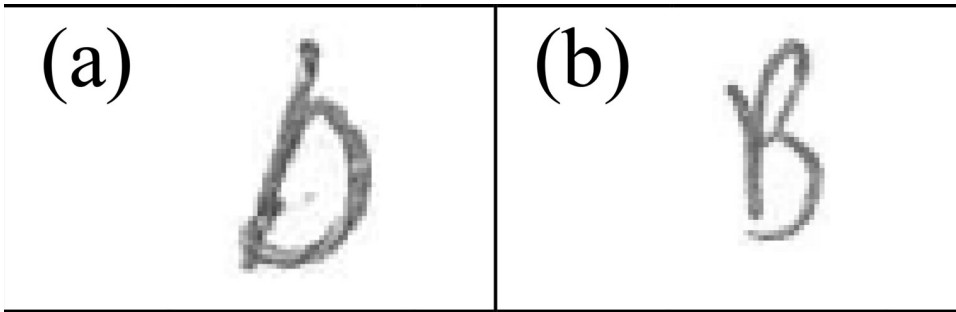

**Fig 3. Characteristic handwritten allographic features from writers from National schools.**

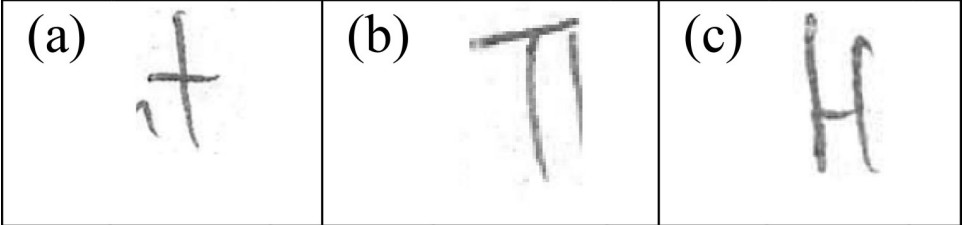

**Fig 4. Characteristic handwritten allographic features from writers from Chinese vernacular schools.**

this current study, could also be due to the characteristics of Arabic script which connects one letter to another in a word.

English handwriting writers from Islamic religious school background were also characterized by the construction of respective letters "B" [Fig 2(D)], "D" [Fig 2(E)], and "p" [Fig 2(F)]. The construction of these letters is significantly different from handwritings produced by Chinese and Tamil medium vernacular school backgrounds individuals. As the Arabic scripts are mostly cursive, and the letters are usually connected, writers in Group D preferred to construct these letters in a single stroke. In other words, writers in Group B and Group C were more likely to form the letters "B", "D", and "p" with at least two individual strokes with the presence of at least one pen lift. There was no significant difference between writers in Group A and Group D for these three letters as they could have also learned Arabic language and Jawi—a writing with very similar formation in Arabic writing during formal schooling.

Significantly high frequencies of lowercase "w" with sharp left and right vertices [Fig 2(G)], as well as left and right sections apart [Fig 2(H)]; were evident in handwriting of individuals from Group D compared to those from Group A. Distinct and pointed vertices of the "w" had discriminated these two groups. Although Arabic script runs from right to left, it seems that there was no significant impact being imparted on the direction of letter formation in English handwriting of the writers from both groups. Clockwise or anti-clockwise direction in letter construction, such as "O" and "Q" were also reported to be not discriminative based on the examination of handwriting samples from these two groups.

## Handwritten allographic features of National school background

Uppercase "D" with protruding initial stroke [Fig 3(A)] was found to be a characteristic feature in English handwriting of individuals from Group A. The writers of Group A also preferred to construct uppercase "B" with protruding initial stroke [Fig 3(B)], compared to handwriting from Group C writers who prominently construct "B" without the protruding initial stroke. This indicates that writers from the National school background preferred writing "B" and "D" with the second stroke extending from the base of the letter from the end of the first stroke instead of starting from the cap-height. Consequently, such writing habits led to the formation of protruding branch-like stroke at the initial start of the letters.

## Handwritten allographic features of Chinese vernacular school background

While constructing a Chinese character with both horizontal and vertical strokes, writers are more likely to make the former followed by the latter in sequence. This was evident in the construction of letter "t" [Fig 4(A)] by Group B writers. This feature was also found to be the most prominently discriminating feature with the highest discriminating power. Letter "T" [Fig 4(B)] also demonstrated the preference of Group B writers to construct the character starting with the arm first followed by the vertical stroke, differentiating them from Group D writers,

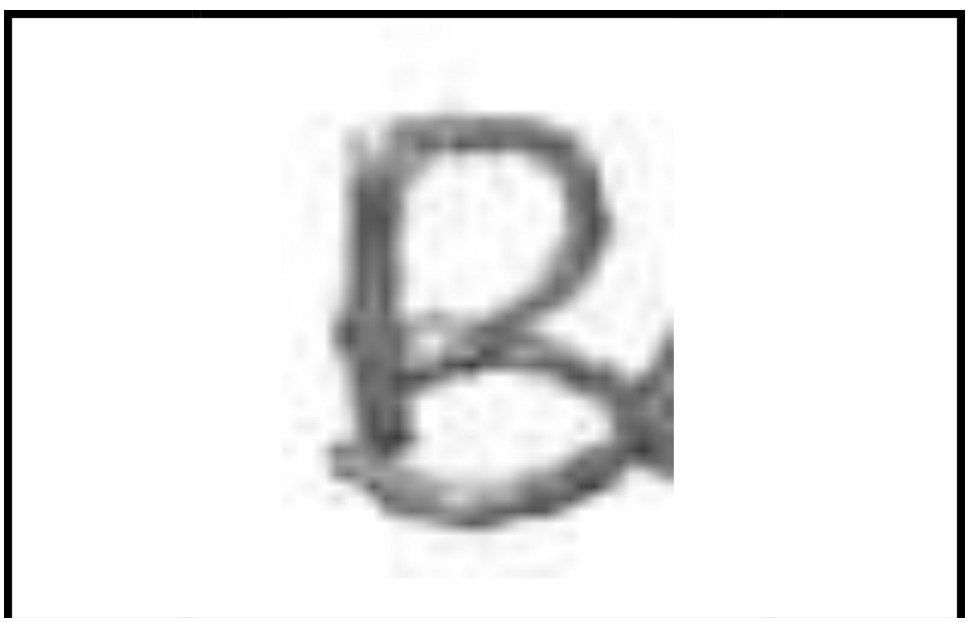

**Fig 5. The only handwritten allographic feature from writers from Tamil vernacular schools.**

who prefer writing the other way. It was important to note that this feature concerning letter "T" did not appear distinguishable between the individuals from Groups A, B, and C. However, Cheng et al. [10] reported that this feature was found to be one of the class characteristics of Chinese writers, making their handwriting distinguishable from the handwriting of Malay and Indian writers.

The basic rule in forming a Chinese character is through the sequence of strokes from the top to the bottom, and from the left to the right. The manner of stroke construction in writing Chinese characters had in a way dominated and influenced the handwriting of English letters, including letter "H" [Fig 4(C)] in this present study. Writers from the Chinese vernacular school were found more likely to construct the uppercase "H" by forming the commanding vertical stroke from top to bottom first, followed by the horizontal crossbar from left to right, and lastly a vertical stroke to complete the letter.

## Handwritten allographic features of Tamil vernacular school background

Individuals from Tamil vernacular school use Tamil language as the medium for learning. Construction of characters in the Tamil writing system involves various combinations of vowel and consonant shapes. Strokes of Tamil syllables are generally curved and rounded. Some syllables have curls and loops at its terminal end. Additionally, the letters appear individually, and they are not cursive or connected to its adjacent letter as demonstrated in Arabic writing system. These letters are also written in the direction from left to right as in English writing.

In this study, it was found that only one allographic feature, namely the letter "B" with no protruding initial stroke (Fig 5), appeared as a characteristic in Group C writers that differentiated them from Group D writers. In the latter group, writers were more likely to generate this letter with a protruding initial stroke. Although other letters such as "D", "P", and "p" have similar stroke numbers and sequence as "B", the absence of protruding initial stroke did not appear consistently in these letters.

Cheng et al. [10] reported that rounded apex of letter "A" and rounded vertex of letter "W" and "V" were exhibited in the handwriting of Indians. However, we have observed otherwise in this study. The design of apex of letter "A", whether it is sharp or rounded, was not seen to be a feature which could distinguish between the groups herein. The said feature seems to randomly occur in the handwriting of all the writers regardless of their schooling background. Shah and Dahiya [6] found that rounded apex of letter "A" was an attribute observed among the Hindi writers in India instead of Tamilians. Most Indians in Malaysia are Tamilians and almost all writers who attend Tamil medium vernacular school do not have Hindi language knowledge. Letter "w" with rounded vertices were found significant to differentiate handwriting samples of Group C individuals from writers of Group D but not for the other two groups. In fact, this study observed that allographic feature representing Indian vernacular school characteristics was very limited.

This study adopted the detailed examination of individual allographs on a subset of 30 handwriting samples from four different school systems to identify if there is any characteristic and discriminative allographic features amongst writers. Based on the frequency of occurrence as the indicative of handwriting characteristics in a study group, 13 allographic features were identified. The number of features identified were greater than the six class characteristics identified by Cheng et al. [10] in the English handwriting of Singaporeans in a similar study. However, it is worth noting that Singapore has a limited language choice education system compared to the multi educational system in the current study.

Handwritings of Islamic religious school background individuals were found to have carried more unique characteristics, particularly in letters "A", "B", "D", "p", "w", "X" and "x". Writers from Chinese vernacular schools exhibited special features in letters "H", "T" and "t" as discriminative characteristics, especially the lowercase "t" which was observed in more than 80% of the handwriting samples from Group B and less than 14% in handwriting samples of other groups. Handwritings of individuals from National schools and Indian vernacular schools had less discriminative allographic features in their respective handwriting samples. Only "B" and "D", and "B" and "w" showed some discriminative characteristics in Group A and Group C, respectively.

In this study, the 13 allographic features revealed in Malaysians' handwriting could serve as indicators of their primary educational background. Along with mother-tongue language learnt in the respective vernacular schools and Islamic religious schools, an individual also learns English and Malay languages as compulsory languages in Malaysian schools. Exposure to these languages may also influence one's style of writing and result in specific handwriting characteristics. The findings herein suggest that the potential influence of taught-writing style and copybook systems in the English handwritings of writers, and that the formation of letters could be strongly associated to an individual's primary educational background. It is also possible for forensic document examiners to consider these allographic features in giving opinions during handwriting examination and to provide insight into the likely identity of the writer, especially when a reference sample is not available [11, 15].

By further including the multinomial logistic regression model in this study, it allowed for the understanding of which primary education system has a higher probability of shaping specific handwritten allographic features, adjusted with confounding variables. Variations in school backgrounds, or more specifically, the unique learning systems reported herein were believed to have influenced the habitual formation of letters during writing [1, 2, 16].

Previous studies had suggested foreign influence toward the formation of handwriting, particularly in countries that have welcomed a huge number of immigrants in the past [4–7, 17]. This study had further proposed that even when individuals are from the same country, their handwriting could be differed, probably due to their respective taught writing styles. It was

important to note that school background is not the only factor underlying handwriting style, as other factors such as the adequacy of standards [1, 2], health impairment [1, 2, 18–20], the influence of drugs and alcohol [1, 2, 21–23], as well as the concentration level during the writing procedure [1, 2] might also contribute to differences in writing. In this study, only writers who fulfilled the inclusion and exclusion criteria were recruited to minimize the effect of above-mentioned potential factors.

Since this study involved only the validated data as examined by the research team including a certified senior forensic document examiner, the inter-rater reliability was not investigated. Based on the identified features found in the current study, a greater number of observers coding the same handwriting samples is proposed to determine the inter-rater reliability among different individuals in future studies. It is also worth exploring the reliability between experienced examiners and individuals with no prior handwriting analysis experience to establish the validity of handwritten allographic features in grouping the writers.

The presence of handwriting allographic features common to a group of writers can be further expanded to other parameters, including gender, ethnicity, age group, handedness and any other relevant information contributing to forensic intelligence. Serving as an exploratory study, the examination of handwritten allographic features was restricted to 30 writers of limited specimens from each group of study. It is acknowledged that a greater population of handwriting samples could further explore the influence of copybook system and taught writing style during writers' primary education.

Our on-going research on a larger population size would certainly allow for determination of allographic features in the handwriting of different groups coupled with analysis of each characteristic based on educational learning systems, with the hope that it will enable the comparison of different handwriting profiles for better forensic intelligence. The collation of information on a source or an author coming from a particular group or background education system through a forensic intelligence framework would be beneficial to the investigation of document related disputes and for use in criminal or civil court trials.

## Conclusion

Learned writing systems is one of the factors that contribute to one's writing style. This study investigated the common handwritten allographic features in English handwritings of writers from different primary education backgrounds. 13 allographic features were identified and they could be attributed to a specific primary educational background the writer had undergone. In routine handwriting examination, the comparison of a questioned sample with reference samples from a known writer is frequently conducted with the aims to identify the author or to detect a forged signature. However, there are instances where comparable writing samples are not available, commonly due to inability to trace a suspect. Nonetheless, the examination of questioned handwriting can still be expanded to retrieve useful information thus narrowing down the possible groups of potential writers, in this case, providing information on the educational backgrounds. It is hoped that determination of these features can then promote the handwriting examination to a wider application to trace or link a source or person for both criminal and civil forensic document investigation, especially when countries are populated with people from different educational backgrounds due to increases in cross border activities and migrations.

## Supporting information

**S1 Data.**
(XLSX)

## Author Contributions

**Conceptualization:** Kah Haw Chang, Ahmad Fahmi Lim Abdullah.

**Data curation:** Linthini Gannetion, Poh Ying Lim.

**Formal analysis:** Kong Yong Wong, Poh Ying Lim, Kah Haw Chang.

**Funding acquisition:** Ahmad Fahmi Lim Abdullah.

**Investigation:** Linthini Gannetion, Kong Yong Wong.

**Methodology:** Linthini Gannetion, Kah Haw Chang, Ahmad Fahmi Lim Abdullah.

**Resources:** Ahmad Fahmi Lim Abdullah.

**Supervision:** Ahmad Fahmi Lim Abdullah.

**Validation:** Kong Yong Wong, Poh Ying Lim.

**Writing – original draft:** Linthini Gannetion.

**Writing – review & editing:** Kong Yong Wong, Poh Ying Lim, Kah Haw Chang, Ahmad Fahmi Lim Abdullah.

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
