## [Decision Letter · Decision Letter 0]

9 Sep 2021

PONE-D-20-35443

An exploratory study on the handwritten allographic features of multi-racial population with different educational backgrounds

PLOS ONE

Dear Dr. Abdullah,

Thank you for submitting your manuscript to PLOS ONE. After careful consideration, we feel that it has merit but does not fully meet PLOS ONE’s publication criteria as it currently stands. Therefore, we invite you to submit a revised version of the manuscript that addresses the points raised during the review process.

I would like to sincerely apologize for the delay you have incurred with your submission. It has been exceptionally difficult to secure reviewers to evaluate your study. We have now received five completed reviews; their comments are available below. The reviewers have raised significant scientific concerns about the study that need to be addressed in a revision.

Please revise the manuscript to address all the reviewer's comments in a point-by-point response in order to ensure it is meeting the journal's publication criteria. Please note that the revised manuscript will need to undergo further review, we thus cannot at this point anticipate the outcome of the evaluation process.

We look forward to receiving your revised manuscript.

Kind regards,

Miquel Vall-llosera Camps

Senior Editor

PLOS ONE

Journal Requirements:

Reviewers' comments:

Reviewer's Responses to Questions

**Comments to the Author**

1. Is the manuscript technically sound, and do the data support the conclusions?

Reviewer #1: Yes

Reviewer #2: Partly

Reviewer #3: Partly

Reviewer #4: Partly

Reviewer #5: Yes

2. Has the statistical analysis been performed appropriately and rigorously? 

Reviewer #1: Yes

Reviewer #2: I Don't Know

Reviewer #3: Yes

Reviewer #4: No

Reviewer #5: Yes

3. Have the authors made all data underlying the findings in their manuscript fully available?

Reviewer #1: Yes

Reviewer #2: No

Reviewer #3: No

Reviewer #4: No

Reviewer #5: No

4. Is the manuscript presented in an intelligible fashion and written in standard English?

Reviewer #1: Yes

Reviewer #2: Yes

Reviewer #3: Yes

Reviewer #4: Yes

Reviewer #5: Yes

5. Review Comments to the Author

Reviewer #1: Writing styles are culturally driven throughout writing learning and teaching. Graphic traditions for a singe writing system may push learners and writers toward common attitude and behavior in reproducing similar styles. The study is very much oriented toward this kind of consideration on individual variation; I would recommend to highlight better relevant applications of the authors' results.

Reviewer #2: General comments

1. In the introduction, the Rationale could be emphasized since the reader could not have any clue understanding of why it is important to study this question and Plos one is not a journal specialized in forensics. The discussion is clearer about the purpose for the study e.g. l300-301

2. The authors mentioned that “All relevant data are within the manuscript. “However, raw data and analysis scripts are not available? Why? Would it possible to publish them in Open Science framework for instance? That would improve replicability and transparency of this work, e.g. comparing with text written in other countries

e.g. Drotár, P., & Dobeš, M. (2020). Dysgraphia detection through machine learning. Scientific reports, 10(1), 1-11.

3. The methodology of selection of letters allographs is unclear “Based on the handwriting samples, eighteen allographic features were suggested as the characteristics which potentially occurred in one group as compared to other groups, covering 14 letters”

4. Were Level of education, sex and handedness data collected and could they

be confounding variables? Were the groups well balanced concerning these variables? Consider presenting a (demographic) table

e.g. Gargot, T., Asselborn, T., Pellerin, H., Zammouri, I., M. Anzalone, S., Casteran, L., ... & Jolly, C. (2020). Acquisition of handwriting in children with and without dysgraphia: A computational approach. PLoS One, 15(9), e0237575

5. What could be the role of new technology? Using scan or electronic tablets?

e.g. Yogarajah, P., & Bhushan, B. (2020, November). Deep Learning Approach to Automated Detection of Dyslexia-Dysgraphia. In The 25th IEEE International Conference on Pattern Recognition.

Or

Asselborn, T., Gargot, T., Kidziński, Ł., Johal, W., Cohen, D., Jolly, C., & Dillenbourg, P. (2018). Automated human-level diagnosis of dysgraphia using a consumer tablet. NPJ digital medicine, 1(1), 1-9.

6. L120 Results and discussions should be separated

7. Would it be possible to identidy the race of a writer from features identified in this paper? Would this method reliable? Would it be possible to compute a sensibility/specificity and intraclass correlation between raters? Would it be next steps and how?

Specific comments

8. L109 All identified handwritten allographic. How many were not identified? How many subjects were approached and eligible? Consider a flow chart. See for instance https://www.equator-network.org/reporting-guidelines/stard/

9. A p-value <0.05 was considered statistically significant. The methodology of such study could lead to false positives. Did authors have any priory hypothesis? Did they pre-registered them? Did they consider a Bonferroni correction?

10. Table 2. Tables should be readable alone. Consider describing the groups instead of calling them A,B,C and D

11. The authors did not report any limitation paragraph in their discussion

Typos :

Abstract and l8 : “to selected population” : not adding anything

L 5. “in the country of this study” - >in Malaysia

L 106. there was error-free ? ->were error-free.

L 161. From the results,” doesn’t had anything”

l 19 “Litterature suggest” doesn’t had anything.

L138. « was demonstrated » -> was reported

L285 “were found to have possessed“ -> had

L21 : in western countries

L91. Dpi : Dot per inch should be also explained in full text

Reviewer #3: The manuscript is interesting and of value in the context of forensic science. However, some revisions are needed:

1. The examination of handwritten samples: it is unclear who performed this examination? Forensic experts? or people without experience? Then, how it was performed and how it was coded; more information on procedure is needed.

2. There is no information on age, sex of participants, please report M and SD. How reading and writing capacities of participants were controlled/tested? Is it is possible that they have any visual and motor impairments? Is this was controlled? It is possible that they differ in intelligence level, is it was controlled? And how?

3. There is no limitation of the presented study. Please describe the limitations, it seems they are several.

Reviewer #4: The overall manuscript is readable and can be understood with ease. It though cannot hold on to the readers attention due to simple write up of results.

More comparative studies in the area can be citied so as to develop the introduction of the paper.

Some studies that can be looked at are:

1. van der Plaats, R.E., van Galen, G.P.

9534939400;57200608568;

Allographic variability in adult handwriting

(1991) Human Movement Science, 10 (2-3), pp. 291-300. Cited 6 times.

https://www.scopus.com/inward/record.uri?eid=2-s2.0-28144445689&doi=10.1016%2f0167-9457%2891%2990008-L&partnerID=40&md5=1c9c01fffdfaab36f1d7ebd1e49c5f05

DOI: 10.1016/0167-9457(91)90008-L

AFFILIATIONS: NICI, University of Nijmegen, Nijmegen, Netherlands

Demographic details other than the age of respondents (18-20) are missing in the paper.

2. Tan, G.X., Viard-Gaudin, C., Kot, A.C.

35300856200;9133978000;35588578100;

Online writer identification using alphabetic information clustering

(2009) Proceedings of SPIE - The International Society for Optical Engineering, 7247, art. no. 72470F, .

https://www.scopus.com/inward/record.uri?eid=2-s2.0-62249120403&doi=10.1117%2f12.805644&partnerID=40&md5=b4210fff315309017e4b63a0c5def5b8

DOI: 10.1117/12.805644

AFFILIATIONS: Centre for Information Security, Nanyang Technological University of Singapore, SINGAPORE, Singapore;

IRCCyN-UMR CNRS 6597, Ecole Polytechnique de l'Université de Nantes, France

The authors mention a researcher team but do not provide an interrater agreement score, so as to defend the choice of letters used in the study.

The number of samples for each study group does not seem adequate to conclude on the results.

The paper should have a table that shows examples of original handwriting of all four groups for highlighting each letter studied for comparing allographic features.

The conclusion can be more elaborate, may be include some examples of forensic cases wherein such allographic study was useful in identifying the criminal and the importance of such research in the country of study.

The manuscript misses data doi.

The authors can mention opportunities in future studies

Reviewer #5: The experimental design of this paper is rigorous, and there is also a scientific hypothesis test

method to some extent.

There are some problems. First, the article lacks a detailed description of data descriptive analysis,

and has not analyzed the statistical characteristics and distribution of the collected data.

In addition, the amount of data collected is only 120, which is relatively small in the experiment of a

large number of results to be verified. It will inevitably cause some test to show significant due to

small samples, distribution imbalance or other accidental factors.

Finally, the data is collected scientific and detailed, but the analysis method is single, which is a Chi

Square test, and the rest is multi-character analysis description and conclusions, lack of

effectiveness and innovation.

Overall, in terms of methods, the whole paper is not innovative enough. In short, the author just uses

Pearson's chi-square test to analyze the new data set collected.

In addition, some parts are confusing, which also significantly deteriorate the clarity and readability

of the paper:

1.Pearson’s chi-square test is used in the Statistical analysis section of this paper. However, the

expression of this method is not clear enough, and there is no detailed mathematical formula

introduction and citation description. For those who are not familiar with Pearson's chi-square test,

it is difficult to understand this method.

2.In table 2, it is not explained how the frequency characteristics are derived;

3.There is no explaination about what Group A, Group B, Group C and Group D represent in table 2；

4.The results for multiple comparison are introduced inTable 3, but there is no specific introduction

on how to perform multiple comparisons.

6. PLOS authors have the option to publish the peer review history of their article (what does this mean?). If published, this will include your full peer review and any attached files.

Reviewer #1: **Yes: **Prof. Barbara Turchetta

Reviewer #2: **Yes: **Thomas Gargot

Reviewer #3: No

Reviewer #4: **Yes: **Sheetal Thomas

Reviewer #5: No

---

## [Author Response · Author response to Decision Letter 0]

12 Oct 2021

Dear Editors and Reviewers of PLOS ONE,

Thank you for your kind review and thoughtful comments. We appreciate the constructive feedback and have carefully taken the comments into consideration in preparing this manuscript. The following summarised our response to the comments:

Reviewer #1

Comment

Writing styles are culturally driven throughout writing learning and teaching. Graphic traditions for a single writing system may push learners and writers toward common attitude and behaviour in reproducing similar styles. The study is very much oriented toward this kind of consideration on individual variation; I would recommend to highlight better relevant applications of the authors' results.

Response

Thank you for the comment. The rationale and relevant application of the current study was included in the introduction (Line 55 – 68). In brief, handwriting examination frequently involves the comparison of an unknown writing sample with samples from a known writer. However, there were instances where suspect or victim is unknown or in the absence of comparable handwriting samples, that makes the application of conventional QDE methodology using known and suspect samples become impossible. Our study revealed that, through the examination of questioned handwriting, it could help in narrowing down the possible groups of potential writers and providing further information in document related fraud. 

Reviewer #2

Comment 1. 

In the introduction, the Rationale could be emphasized since the reader could not have any clue understanding of why it is important to study this question and Plos one is not a journal specialized in forensics. The discussion is clearer about the purpose for the study e.g. l300-301 

Response 1. 

Thank you for the suggestion. The rationale and importance of the study was included in the manuscript (Line 55 – 68). Although Plos One is not a specialised journal in forensic science; however, this study is transdisciplinary in nature where it could be explored from the perspectives of human sciences and digital sciences.

Comment 2. 

The authors mentioned that “All relevant data are within the manuscript. “However, raw data and analysis scripts are not available? Why? Would it possible to publish them in Open Science framework for instance? That would improve replicability and transparency of this work, e.g. comparing with text written in other countries

e.g. Drotár, P., & Dobeš, M. (2020). Dysgraphia detection through machine learning. Scientific reports, 10(1), 1-11. 

Response 2. 

Thank you. For reference purpose, our raw data in the form of Excel file is supplied as supplementary file.

Comment 3. 

The methodology of selection of letters allographs is unclear “Based on the handwriting samples, eighteen allographic features were suggested as the characteristics which potentially occurred in one group as compared to other groups, covering 14 letters” 

Response 3.

Thank you for the comments. The procedure of examination and selection of the eighteen allographic features was described in the revised manuscript (Line 153-160).

Comment 4. 

Were Level of education, sex and handedness data collected and could they be confounding variables? Were the groups well balanced concerning these variables? Consider presenting a (demographic) table

e.g. Gargot, T., Asselborn, T., Pellerin, H., Zammouri, I., M. Anzalone, S., Casteran, L., ... & Jolly, C. (2020). Acquisition of handwriting in children with and without dysgraphia: A computational approach. PLoS One, 15(9), e0237575 

Response 4. 

Thanks for suggestion. We have included the table describing the demographic factors (Table 2) and tested the differences among the groups. Those significant variables were adjusted as confounding variables in the final model.

Comment 5. 

What could be the role of new technology? Using scan or electronic tablets?

e.g. Yogarajah, P., & Bhushan, B. (2020, November). Deep Learning Approach to Automated Detection of Dyslexia-Dysgraphia. In The 25th IEEE International Conference on Pattern Recognition.

Or

Asselborn, T., Gargot, T., Kidziński, Ł., Johal, W., Cohen, D., Jolly, C., & Dillenbourg, P. (2018). Automated human-level diagnosis of dysgraphia using a consumer tablet. NPJ digital medicine, 1(1), 1-9. 

Response 5. 

Thank you for the suggestion. The findings from this study would provide us the understanding on handwritten pattern among Malaysians. Subsequently, the collation of information on a source or an author coming from a particular group or background education system through a forensic intelligence framework would be beneficial to the investigation of document related cases. Additional information was included as future recommendation in the revised manuscript (Line 397-401, 412-414). 

Comment 6. 

L120 Results and discussions should be separated 

Response 6. 

As suggested by the reviewer, Results and Discussion were separated into two sections. 

Comment 7. 

Would it be possible to identify the race of a writer from features identified in this paper? Would this method reliable? Would it be possible to compute a sensibility/specificity and intraclass correlation between raters? Would it be next steps and how? 

Response 7. 

In this study, the race of a writer was demonstrated in Table 2. It is noted that majority of the writers from the respective school group is belong to one race group. For instance, 100% of respondents from Chinese and Tamil schools were non-Malay. Therefore, this variable was not tested for its possibility for discrimination. In this study, only two confounding variables (age and secondary schooling system) were adjusted in the final model (Line 174-177).

Procedure of the examination and coding of the handwriting features was described in the revised manuscript (Line 153 – 160). In brief, the handwriting samples were examined in two different occasions by two teams of researcher independently, and their results were validated by a certified forensic document examiner. As future recommendation, inter-rater reliability study involving multiple raters examining and coding the same set of handwriting samples shall be conducted, upon determination of specific allographic features useful for discrimination among the groups (Line 397-401). 

Comment 8. 

L109 All identified handwritten allographic. How many were not identified? How many subjects were approached and eligible? Consider a flow chart. See for instance https://www.equator-network.org/reporting-guidelines/stard/

Response 8. 

The sampling method and subject recruitment of this study was included in the revised manuscript (Line 80-86). This study was aimed to identify and establish the characteristic handwritten allographic features corresponding to selected population from different primary educational backgrounds. 

Comment 9. 

A p-value <0.05 was considered statistically significant. The methodology of such study could lead to false positives. Did authors have any priory hypothesis? Did they pre-registered them? Did they consider a Bonferroni correction? 

Response 9. 

Thank you for the comments. We have included multinomial logistic regression model, adjusted with confounding variables in the revised manuscript. Our aim of the study is to investigate the probability of respondents of having that handwriting features are more likely from which particular group.

Comment 10. 

Table 2. Tables should be readable alone. Consider describing the groups instead of calling them A,B,C and D 

Response 10. 

Thank you for the suggestion. The groups were named according to their educational systems (Table 3 in the revised manuscript). 

Comment 11. 

The authors did not report any limitation paragraph in their discussion 

Response 11. 

Thank you for the suggestion. A separate section on limitations was included in the revised manuscript (Line 397 –408). 

Comment 12. 

Typos :

Abstract and l8 : “to selected population” : not adding anything

L 5. “in the country of this study” - >in Malaysia

L 106. there was error-free ? ->were error-free.

L 161. From the results,” doesn’t had anything”

l 19 “Litterature suggest” doesn’t had anything.

L138. « was demonstrated » -> was reported

L285 “were found to have possessed“ -> had

L21 : in western countries

L91. Dpi : Dot per inch should be also explained in full text 

Response 12. 

Thank you for the comments. Grammatical and typo errors were corrected accordingly.

Reviewer #3

Comment. 

The manuscript is interesting and of value in the context of forensic science. However, some revisions are needed:

Response. 

Thank you.

Comment 1. 

The examination of handwritten samples: it is unclear who performed this examination? Forensic experts? or people without experience? Then, how it was performed and how it was coded; more information on procedure is needed.

Response 1. 

Thank you for the suggestion. The procedure of handwriting examination was included in the revised manuscript (Line 153-160).

Comment 2. 

There is no information on age, sex of participants, please report M and SD. How reading and writing capacities of participants were controlled/tested? Is it is possible that they have any visual and motor impairments? Is this was controlled? It is possible that they differ in intelligence level, is it was controlled? And how?

Response 2. 

Thank you for the suggestion. The information on age, gender and other demographic data was included as described in Table 2. 

The inclusion criteria of the writers who had participated in this study was described (Line 90-93). In general, they were university students who are mentally challenged, no physical disabilities and well-versed in reading and writing in English Language. 

Comment 3. 

There is no limitation of the presented study. Please describe the limitations, it seems they are several.

Response 3. 

Thank you for the suggestion. Limitations of the study were included in the revised manuscript (Line 397 –408). 

Reviewer #4

Comment 1.

The overall manuscript is readable and can be understood with ease. It though cannot hold on to the readers attention due to simple write up of results. 

Response 1. 

Thank you. The manuscript is revised to include additional figures and table for the better understanding of readers.

Comment 2.

More comparative studies in the area can be citied so as to develop the introduction of the paper.

Some studies that can be looked at are:

1. van der Plaats, R.E., van Galen, G.P.

9534939400;57200608568;

Allographic variability in adult handwriting

(1991) Human Movement Science, 10 (2-3), pp. 291-300. Cited 6 times.

2. Tan, G.X., Viard-Gaudin, C., Kot, A.C.

35300856200;9133978000; 35588578100;

Online writer identification using alphabetic information clustering

(2009) Proceedings of SPIE - The International Society for Optical Engineering, 7247, art. no. 72470F

Response 2. 

The introduction part was revised where the rationale and relevant application of the current study was added. Thank you for the suggestions on articles; however, the suggested articles might not be fitted into the current study as it more focused on the influence of educational backgrounds. The suggested articles shall be considered in our subsequent study which aims on individualizing the allographic features and establishing the algorithm model for discrimination.

Comment 3.

The authors mention a researcher team but do not provide an interrater agreement score, so as to defend the choice of letters used in the study. 

Response 3. 

Thank you for the suggestion. The description on the examination and coding of allographic features was included in the revised manuscript (Line 153-160). An inter-rater reliability study shall be proposed, perhaps to involve expert and non-experts to investigate the reliability of the method in the current study as a future recommendation in the revised manuscript (Line 398-401). 

Comment 4.

The number of samples for each study group does not seem adequate to conclude on the results.

Response 4. 

Thank you. This study serves as the exploratory study to identify the allographic features for the discrimination among the writers of different educational backgrounds. In the revised manuscript, the power of study was also tested where majority of the identified features achieved more than 78% of power. The number of writers in the current study was also included as one of the limitations of this study, which deserving further investigation in the future with greater number of participants. 

Comment 5.

The paper should have a table that shows examples of original handwriting of all four groups for highlighting each letter studied for comparing allographic features. 

Response 5. 

Thank you for the suggestion. Table 1 in the manuscript describes the allographic features used to compare the handwriting samples among the writers of different educational backgrounds. To provide better understanding, images of the characteristic allographic features identified in this study from the writers of different educational backgrounds were included in the revised manuscript (Figure 1-4). 

Comment 6.

The conclusion can be more elaborate, may be include some examples of forensic cases wherein such allographic study was useful in identifying the criminal and the importance of such research in the country of study. 

Response 6. 

Thank you for the suggestion. The examples of forensic cases wherein such determination is important was included in the introduction (Line 55-68). The conclusion was also revised to include the usefulness of data gathered from the current study for forensic science application (Line 421-426). 

Comment 7.

The manuscript misses data doi. 

Response 7. 

The data is submitted as supplementary file. 

Comment 8.

The authors can mention opportunities in future studies 

Response 8. 

Thank you for the suggestion. Future recommendation subsequent to the current study was included (Line 398-401, 404-408).

Reviewer #5

Comment 1. 

The experimental design of this paper is rigorous, and there is also a scientific hypothesis test method to some extent.

Response 1. 

Thank you for the comments. The manuscript was revised based on the comments below. 

Comment 2. 

There are some problems. First, the article lacks a detailed description of data descriptive analysis, and has not analysed the statistical characteristics and distribution of the collected data.

Response 2. 

Thank you for the suggestion. Data descriptive analysis was included in the manuscript (Table 2). The statistical characteristics and distribution were also analysed, and the confounding variables were considered in the subsequent analysis. 

Comment 3. 

In addition, the amount of data collected is only 120, which is relatively small in the experiment of a large number of results to be verified. It will inevitably cause some test to show significant due to small samples, distribution imbalance or other accidental factors.

Response 3. 

Thank you for the comments. Post hoc power analysis to check the sample size was included in the current study, and the results showed that the study carried adequate power to detect the statistically significant difference. The power of this study for each significant variables ranges from 78% to 100%, which indicated that our study has enough power to detect statistically significant (Line 235-239). 

Comment 4. 

Finally, the data is collected scientific and detailed, but the analysis method is single, which is a Chi Square test, and the rest is multi-character analysis description and conclusions, lack of effectiveness and innovation.

Response 4. 

Thank you for the suggestion. In addition to the Chi Square, a multinomial logistic regression analysis was conducted, and the results was adjusted with all possible confounding variables. 

Comment 5. 

Overall, in terms of methods, the whole paper is not innovative enough. In short, the author just uses Pearson's chi-square test to analyse the new data set collected.

Response 5. 

Thank you. A multinomial logistic regression model was proposed in the revised manuscript.

Comment 6. 

In addition, some parts are confusing, which also significantly deteriorate the clarity and readability of the paper:

1.Pearson’s chi-square test is used in the Statistical analysis section of this paper. However, the expression of this method is not clear enough, and there is no detailed mathematical formula introduction and citation description. For those who are not familiar with Pearson's chi-square test, it is difficult to understand this method.

Response 6. 

Thank you for the comments. Subsequent to previous comment, a multinomial logistic regression analysis was carried out. Description on the establishment of the model was included in the methodology (Line 138-149). 

Comment 7. 

In table 2, it is not explained how the frequency characteristics are derived.

Response 7. 

Table 2 was revised (Table 3 in the revised manuscript). 

Comment 8. 

There is no explanation about what Group A, Group B, Group C and Group D represent in table 2.

Response 8. 

Thank you for the suggestion. The groups were named according to their educational systems (Table 3 in the revised manuscript). 

Comment 9. 

The results for multiple comparison are introduced in Table 3, but there is no specific introduction on how to perform multiple comparisons.

Response 9. 

Thank you for the suggestion. The table on the multiple comparison was replaced by the results from multinomial logistic regression analysis (Table 4). The detailed on the statistical procedure was also included in the revised manuscript (Line 138-149).

This manuscript is checked thoroughly. We really appreciate the efforts taken by the editor and reviewers to help us improve the manuscript, and we hope that we have addressed all your concerns in this manuscript. 

Thank you. 

Ahmad Fahmi Lim Abdullah, PhD

School of Health Sciences, Universiti Sains Malaysia, 16150 Kubang Kerian, Kelantan, Malaysia. Email: fahmilim@usm.my; Tel: +609-7677596; Fax: +609-7677515.

---

## [Decision Letter · Decision Letter 1]

1 Nov 2021

PONE-D-20-35443R1An exploratory study on the handwritten allographic features of multi-racial population with different educational backgroundsPLOS ONE

Dear Dr. Abdullah,

Thank you for submitting your manuscript to PLOS ONE. After careful consideration, we feel that it has merit but does not fully meet PLOS ONE’s publication criteria as it currently stands. Therefore, we invite you to submit a revised version of the manuscript that addresses the points raised during the review process.

The manuscript has been further evaluated by three reviewers, and based on their comments, this manuscript is almost ready for publication. However Reviewer #2 still has some outstanding concerns requesting additional methodological details and elaboration in the Discussion section. Could you please revise the manuscript to carefully address the concerns raised?

We look forward to receiving your revised manuscript.

Kind regards,

Avanti Dey, PhD

Staff Editor

PLOS ONE

Journal Requirements:

Reviewers' comments:

Reviewer's Responses to Questions

**Comments to the Author**

1. If the authors have adequately addressed your comments raised in a previous round of review and you feel that this manuscript is now acceptable for publication, you may indicate that here to bypass the “Comments to the Author” section, enter your conflict of interest statement in the “Confidential to Editor” section, and submit your "Accept" recommendation.

Reviewer #2: All comments have been addressed

Reviewer #3: (No Response)

Reviewer #4: All comments have been addressed

2. Is the manuscript technically sound, and do the data support the conclusions?

Reviewer #2: Yes

Reviewer #3: Partly

Reviewer #4: Yes

3. Has the statistical analysis been performed appropriately and rigorously? 

Reviewer #2: Yes

Reviewer #3: Yes

Reviewer #4: Yes

4. Have the authors made all data underlying the findings in their manuscript fully available?

Reviewer #2: Yes

Reviewer #3: Yes

Reviewer #4: Yes

5. Is the manuscript presented in an intelligible fashion and written in standard English?

Reviewer #2: Yes

Reviewer #3: Yes

Reviewer #4: Yes

6. Review Comments to the Author

Reviewer #2: Thank you to your answers to my comments. Thank you all for this revision work. The structure and the scientific content of the article are much better.

Reviewer #3: The authors have improved the manuscript. They inserted the data on examination of handwritten samples and on participants. However there are still some aspects that need revision. Please include the inter-rater reliability between forensic coders. This should be presented in the manuscript. The limitations are still not elaborated. Many factors can differentiate the forms of letter examined by the researchers, not only education and nationality. These factors such as intelligence or perceptive/manual impairments were not controlled in the present study. The researchers should be aware of these potential factors, particularly in the situation while the sample size is small. The list of references is very short. Please explain the impact of these potential factors on handwriting in Discussion and say why these factors were not controlled as well as enrich the literature on this topic.

Reviewer #4: The authors have addressed to all the comments and the paper seems to be fine now. Though the readership is limited at this point of time, it is an interesting article to be published as a base for future direction of research.

7. PLOS authors have the option to publish the peer review history of their article (what does this mean?). If published, this will include your full peer review and any attached files.

Reviewer #2: **Yes: **Thomas Gargot

Reviewer #3: No

Reviewer #4: **Yes: **Sheetal Thomas

---

## [Author Response · Author response to Decision Letter 1]

12 Dec 2021

Dear Editors and Reviewers of PLOS ONE,

Thank you for your kind review and thoughtful comments. We appreciate the constructive feedback and have carefully taken the comments into consideration in preparing this manuscript. The following summarised our response to the comments:

Reviewer #3

Comment 1:

The authors have improved the manuscript. They inserted the data on examination of handwritten samples and on participants.

Response 1:

Thank you.

Comment 2:

However there are still some aspects that need revision. Please include the inter-rater reliability between forensic coders. This should be presented in the manuscript. The limitations are still not elaborated.

Response 2:

Procedure of the examination and coding of the handwriting features was described in the manuscript in which the handwriting samples were examined in two different occasions by two teams of researcher independently, and their results were validated by a certified forensic document examiner. In this study, only the data validated by the document examiner was considered in interpretation and data analysis (Line 161-162). 

As for the inter-rater reliability among the coders as suggested by the reviewer, a greater number of coders shall be proposed to involve multiple raters examining and coding the same set of handwriting samples, and this was not conducted in the scope of the current study. The inter-rater reliability study would be more useful upon the determination of specific allographic features for discrimination among the groups as demonstrated in the current study, perhaps to involve also expert and non-experts as a future recommendation. Such limitation and future recommendation were included in the revised manuscript (Line 407-413). 

Comment 3:

Many factors can differentiate the forms of letter examined by the researchers, not only education and nationality. These factors such as intelligence or perceptive/manual impairments were not controlled in the present study. The researchers should be aware of these potential factors, particularly in the situation while the sample size is small. 

Response 3:

The inclusion and exclusion criteria of the subjects of this study was included in the revised manuscript (Line 92-94). In general, they were university students who are mentally challenged, no physical disabilities and well-versed in reading and writing in English Language. 

The limitation on the sample size was also included in the revised manuscript (Line 240-241, 418-420), deserving further investigation in the future with greater number of participants.

Comment 4:

The list of references is very short. Please explain the impact of these potential factors on handwriting in Discussion and say why these factors were not controlled as well as enrich the literature on this topic.

Response 4:

Additional references with related information were included in the revised manuscript (Line 397-401). 

The potential factors which could affect the development of handwriting were also discussed in the revised manuscript (Line 401-406).

This manuscript is checked thoroughly. We really appreciate the efforts taken by the editor and reviewers to help us improve the manuscript, and we hope that we have addressed all your concerns in this manuscript. 

Thank you. 

Ahmad Fahmi Lim Abdullah, PhD

School of Health Sciences, Universiti Sains Malaysia, 16150 Kubang Kerian, Kelantan, Malaysia. Email: fahmilim@usm.my; Tel: +609-7677596; Fax: +609-7677515.

---

## [Decision Letter · Decision Letter 2]

15 Mar 2022

PONE-D-20-35443R2An exploratory study on the handwritten allographic features of multi-racial population with different educational backgroundsPLOS ONE

Dear Dr. Abdullah,

Thank you for submitting your manuscript to PLOS ONE. After careful consideration, we feel that it has merit but does not fully meet PLOS ONE’s publication criteria as it currently stands. Therefore, we invite you to submit a revised version of the manuscript that addresses the points raised during the review process. While this manuscript is nearly ready for acceptance, we request that you address a few further concerns. Specifically:

1) Several reviewers previously raised methodology concerns, including sample size, statistical analysis, etc. Although this has been addressed in the study as being relevant for an exploratory study, please provide some additional justification for these issues. 

2) There are some outstanding language issues which, if rectified, would strengthen this submission. For example, some information is missing or unclear, e.g., lines 245-247, line 399, as well as others. Therefore, we highly encourage a thorough proof-reading prior to resubmission. 

3) Please also reconsider a rephrasing of your interchangeable use of 'race' and 'nationality' (some guidelines for this may be found here - https://jamanetwork.com/journals/jama/fullarticle/2783090). 

We look forward to receiving your revised manuscript.

Kind regards,

Avanti Dey, PhD

Staff Editor

PLOS ONE

Journal Requirements:

Reviewers' comments:

Reviewer's Responses to Questions

**Comments to the Author**

1. If the authors have adequately addressed your comments raised in a previous round of review and you feel that this manuscript is now acceptable for publication, you may indicate that here to bypass the “Comments to the Author” section, enter your conflict of interest statement in the “Confidential to Editor” section, and submit your "Accept" recommendation.

Reviewer #2: All comments have been addressed

Reviewer #3: All comments have been addressed

Reviewer #4: All comments have been addressed

2. Is the manuscript technically sound, and do the data support the conclusions?

Reviewer #2: Yes

Reviewer #3: Yes

Reviewer #4: Partly

3. Has the statistical analysis been performed appropriately and rigorously? 

Reviewer #2: Yes

Reviewer #3: Yes

Reviewer #4: Yes

4. Have the authors made all data underlying the findings in their manuscript fully available?

Reviewer #2: Yes

Reviewer #3: Yes

Reviewer #4: No

5. Is the manuscript presented in an intelligible fashion and written in standard English?

Reviewer #2: (No Response)

Reviewer #3: Yes

Reviewer #4: Yes

6. Review Comments to the Author

Reviewer #2: Thank you for this work. I already accepted the paper in the previous round. I hope you the best for the publication

Reviewer #3: The authors have satisfactory revised the manuscript. They considered all the suggestions. Thank you for this.

Reviewer #4: The conclusion can include more application areas for the study. The paper is well structured. Research methodology is adequately developed.

7. PLOS authors have the option to publish the peer review history of their article (what does this mean?). If published, this will include your full peer review and any attached files.

Reviewer #2: No

Reviewer #3: No

Reviewer #4: **Yes: **Sheetal Thomas

---

## [Author Response · Author response to Decision Letter 2]

11 Apr 2022

Dear Editors and Reviewers of PLOS ONE,

Thank you for your kind review and thoughtful comments. We appreciate the constructive feedback and have carefully taken the comments into consideration in preparing this manuscript. The following summarised our response to the comments:

Editor 

Comment:

1) Several reviewers previously raised methodology concerns, including sample size, statistical analysis, etc. Although this has been addressed in the study as being relevant for an exploratory study, please provide some additional justification for these issues. 

Response: 

Thank you for the comment. The justification on the sample size determination and choice of statistical analysis was included in the revised manuscript. 

Line 88-91 - “Sample size was calculated through two proportion formula by considering the letter “U” with introductory stroke [4], 25% of the Polish (0% for English) possessed the handwritten feature. With 80% of power study, 5% of risk error, and 10% of non-response rate, 30 writers were collected from each group.”

Line 386-388 - “By further including the multinomial logistic regression model in this study, it allowed for the understanding of which primary education system has a higher probability of shaping specific handwritten allographic features, adjusted with confounding variables.”

Comment:

2) There are some outstanding language issues which, if rectified, would strengthen this submission. For example, some information is missing or unclear, e.g., lines 245-247, line 399, as well as others. Therefore, we highly encourage a thorough proof-reading prior to resubmission. 

Response: 

Thank you for the comments. 

Line 245-247 was revised to include the percentage of the power of study and its description as follows: 

“Handwritten feature with low powder of study (<70%) is suggested for further investigation with greater sample size, in this case the uppercase "A" with three individual strokes. For the current study, only those statistically significant features with adequate power (in this case greater than 78%) are discussed.” (Line 241-244). 

Line 399 was also revised to “Previous studies had suggested foreign influence toward the formation of handwriting, particularly in countries that have welcomed a huge number of immigrants in the past.” (Line 391-392). 

Comment:

3) Please also reconsider a rephrasing of your interchangeable use of 'race' and 'nationality' (some guidelines for this may be found here - https://jamanetwork.com/journals/jama/fullarticle/2783090).

Response: 

Thank you for the comment. In this study, the subjects were from the three main ethnicity groups, and they are from the same country (same nationality). In the revised manuscript, the term “ethnicity” was used in the results and discussion.

Reviewer #4 

Comment:

The conclusion can include more application areas for the study. The paper is well structured. Research methodology is adequately developed.

Response: 

Thank you for the comment. The conclusion was added with “Nonetheless, the examination of questioned handwriting can still be expanded to retrieve useful information thus narrowing down the possible groups of potential writers, in this case, providing information on the educational backgrounds.” to support the application for the study.

This manuscript is checked thoroughly. We really appreciate the efforts taken by the editor and reviewers to help us improve the manuscript, and we hope that we have addressed all your concerns in this manuscript. 

Thank you. 

Ahmad Fahmi Lim Abdullah, PhD

School of Health Sciences, Universiti Sains Malaysia, 16150 Kubang Kerian, Kelantan, Malaysia. Email: fahmilim@usm.my; Tel: +609-7677596; Fax: +609-7677515.

---

## [Editor Report · Decision Letter 3]

9 May 2022

An exploratory study on the handwritten allographic features of multi-ethnic population with different educational backgrounds

PONE-D-20-35443R3

Dear Dr. Abdullah,

We’re pleased to inform you that your manuscript has been judged scientifically suitable for publication and will be formally accepted for publication once it meets all outstanding technical requirements.

Kind regards,

Avanti Dey, PhD

Staff Editor

PLOS ONE
---

## [Editor Report · Acceptance letter]

8 Jun 2022

PONE-D-20-35443R3 

An exploratory study on the handwritten allographic features of multi-ethnic population with different educational backgrounds 

Dear Dr. Abdullah:

I'm pleased to inform you that your manuscript has been deemed suitable for publication in PLOS ONE. Congratulations! Your manuscript is now with our production department. 

Kind regards, 

on behalf of

Dr. Avanti Dey 

Staff Editor

PLOS ONE